# Does the Pathogenic Sequence of Skeletal Muscle Degeneration in Duchenne Muscular Dystrophy Begin and End with Unrestrained Satellite Cell Activation?

**Carl George Carlson †**

Department of Physiology, A.T. Still University, Kirksville, MO 63501, USA; georgecarlson9@gmail.com
† Retired.

**Abstract:** A recent study describing unique effects of myostatin inhibition on a severely dystrophic (mdx) muscle, and independent experiments showing that branched fibers are uniquely sensitive to activity-induced membrane rupture, have led to a new hypothesis of dystrophic pathogenesis. This hypothesis states that the absence of dystrophin directly results in some degree of unrestrained satellite cell activation that is not dependent upon prior fiber injury. The hypothesis further states that dystrophin promotes satellite cell quiescence, and that its absence directly results in a lack of control over the mechanism(s) by which muscle activity regulates satellite cell activation and fiber growth during passive stretch and concentric and eccentric exercise. The ultimate consequence of this lack of control is to produce branched, weak, and fragile fibers that accumulate at a rate dependent upon the history of activation for each dystrophic muscle. The purpose of this opinion paper is to summarize the results in support of this new hypothesis in an attempt to stimulate further research on the regulation of satellite cell activity in dystrophic muscle.

**Keywords:** Duchenne muscular dystrophy; dystrophin; satellite cell activation; myoblast proliferation; myoblast fusion; myostatin inhibition; mdx mouse

## 1. Introduction

The absence of dystrophin is widely believed to result in structural breakdown of the plasma membrane, resulting muscle fiber injury, and repair and regeneration of the injured dystrophic muscle fiber. With the exception of the first step, this sequence is essentially similar to events that occur within the environment of a normal muscle fiber during concentric and eccentric exercise, resistance exercise, or direct muscle injury [1–3]. The repair of injured nondystrophic muscle fibers is mediated by the activation of satellite cells, which are located immediately proximal to the fiber plasma membrane and are held in place by the tight envelope provided by the internal basal lamina [1]. Once activated, the satellite cells proliferate and give rise to differentiated myoblasts which further multiply in situ, ultimately fusing with the fiber plasma membrane to form new myonuclei that contribute to the protein synthesis and internal cell signaling necessary for the maintenance of a normal, healthy, multinucleate adult muscle fiber [2,4,5]. It is generally supposed that the same repair system is operative in dystrophic muscle; only in this case it operates more or less continually (albeit, periodically) to counteract the continual (or periodic) degeneration of dystrophic muscle fibers whose membranes have been rendered fragile by the absence of the protein dystrophin [6]. Progression of the disease is thus characterized as a continual accretion of injured, degenerating muscle fibers, immune-mediated clean up of the fiber remnants, and continuing cycles of satellite cell activation and myoblast proliferation that culminate in the exhaustion of the normal repair mechanisms and ultimate depletion of healthy muscle fibers (Figure 1).

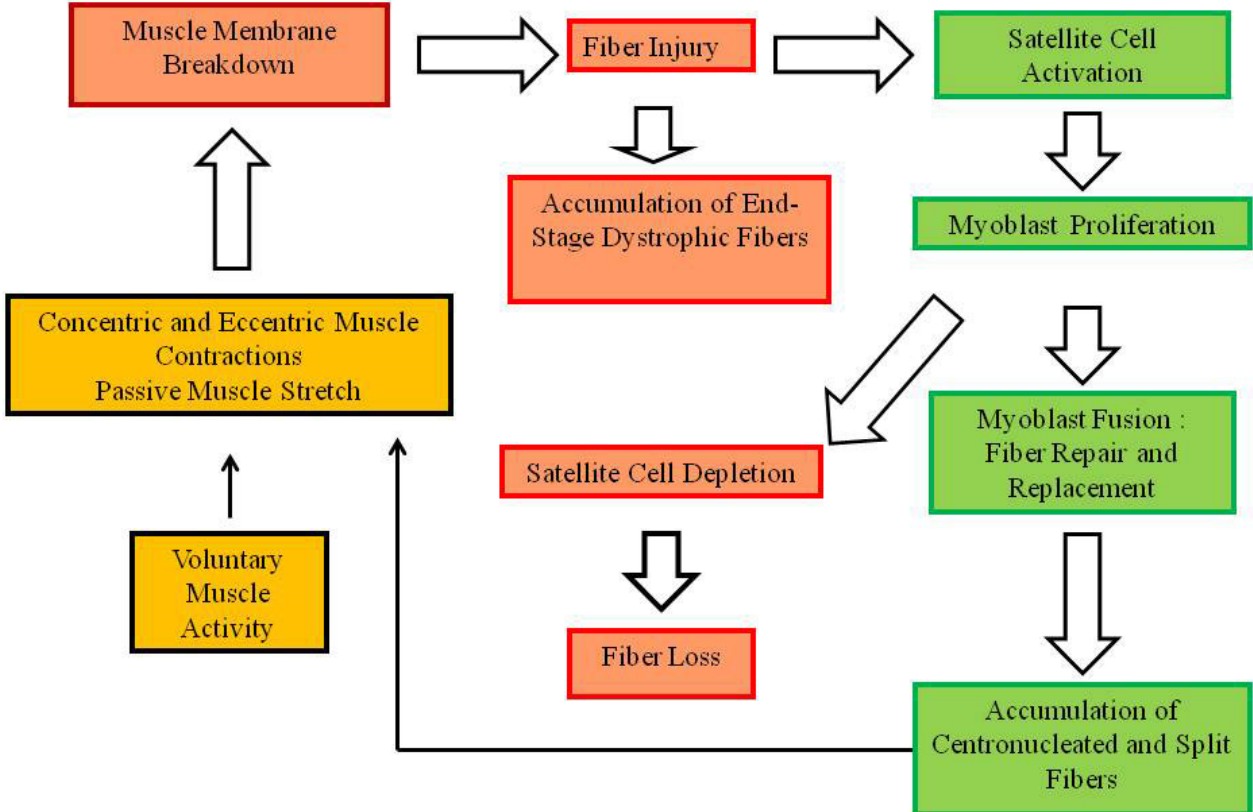

**Figure 1.** Classical pathogenic sequence for Duchenne muscular dystrophy. The sequence is initiated by the breakdown of the muscle plasma membrane during normal voluntary activity. Muscle activities are shown in yellow blocks outlined in black, deleterious consequences of the absence of dystrophin are in light red blocks outlined in red, and compensatory events in green blocks are outlined in green.

An intrinsically limiting characteristic of the scientific method is the fact that the answers to our questions are in part guided by the questions themselves. When dystrophin was first discovered to be absent in Duchenne muscular dystrophy and in the mdx mouse [6], the primary question asked was: "How does the absence of dystrophin lead to muscle degeneration?". A structural similarity of dystrophin to the erythrocyte membrane cytoskeletal protein spectrin [7], and the immunochemical identification of dystrophin as a component of the internal cytoskeleton of skeletal muscle [8] provided circumstantial support to the general idea that Duchenne muscular dystrophy was a "membrane disease". This idea was initially posited prior to the discovery of dystrophin by observations of unusual erythrocyte membrane flexibility in dystrophic patients [9] and electron micrographic studies showing membrane "delta lesions" in Duchenne muscle fibers [10]. Subsequent studies describing the dystrophin–glycoprotein complex as a transmembrane scaffold [11], which could potentially dissipate forces across the membrane during muscle activity, provided an attractive component to this structural model of Duchenne pathogenesis. The idea was thus implanted that the lack of dystrophin produced muscle membrane fragility as the initiating event in a pathogenic cascade leading to progressive necrosis and massive compensatory, but ultimately failed, efforts at muscle regeneration (Figure 1). Although the structural model for dystrophic pathogenesis has been strongly challenged [12], and "delta lesions" [10] have not been observed in several independent studies examining mdx and human Duchenne muscle [13–15], I think it is fair to say that still the most widely held view of the disease is that it stems from a muscle membrane structural defect that leads to secondary activation of normal compensatory regeneration mechanisms that are ultimately

exhausted, thus leading to a total failure to sustain the muscle tension required for the maintenance of life.

Now, considering the diverse pathologies apparent in a typical dystrophic muscle (fiber degeneration, macrophage infiltration, satellite cell activation, myoblast proliferation and fusion, centronucleation, hypercontraction, fibrosis, muscle weakness, split fibers, collagen accumulation etc.), suppose one were to ask the broader question: "How does the absence of dystrophin lead to the dystrophic phenotype?". In this case, one would be led into several diverse hypotheses which would include the structural hypothesis, but would also include hypotheses relating dystrophin to immune function, cell signaling, collagen expression, satellite cell activation, myoblast proliferation, and myoblast fusion with the muscle membrane. The purpose of this opinion paper is to highlight a new proposal to replace the idea that the problem in muscular dystrophy begins with the muscle membrane (Figure 1) with a new idea that the problem instead begins with the satellite cell and its response to normal patterns of muscle activity in an environment lacking dystrophin (Figure 2).

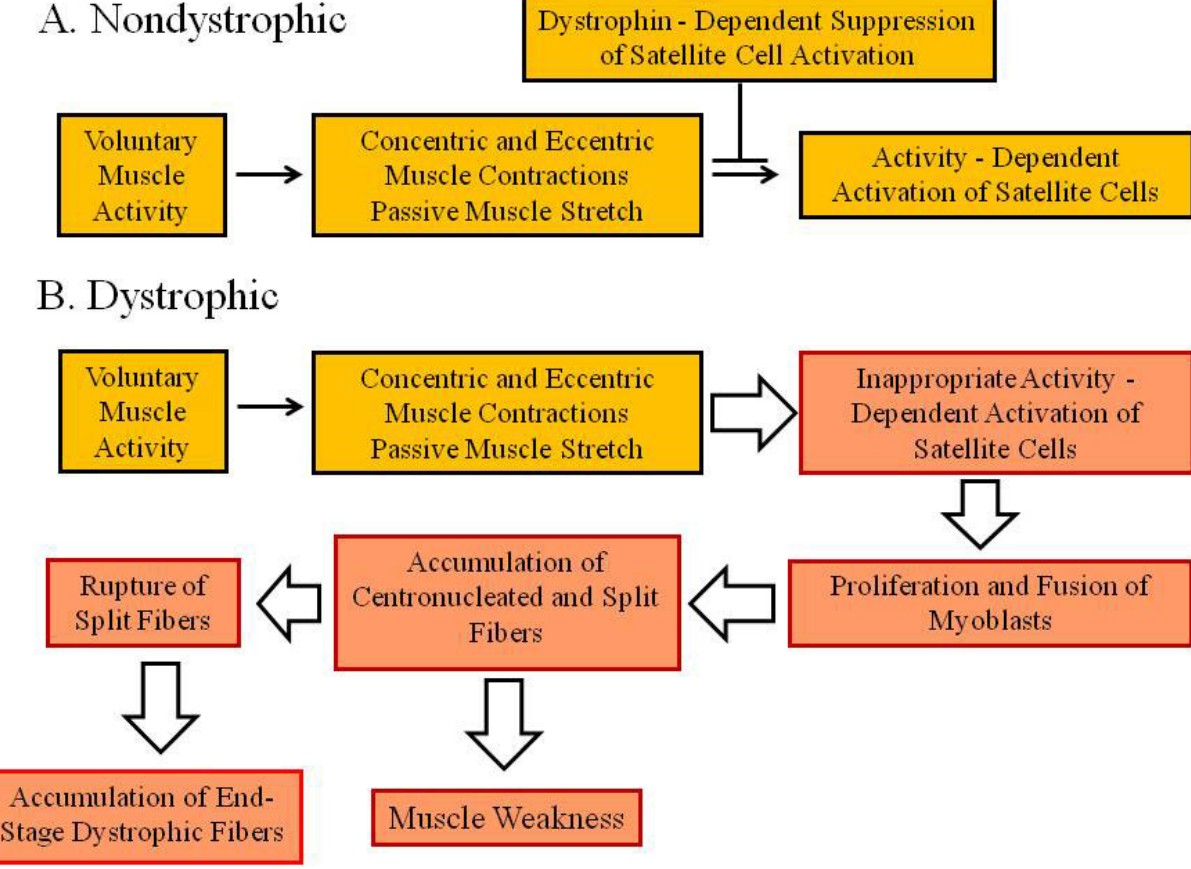

**Figure 2.** A pathogenic sequence for Duchenne muscular dystrophy that is initiated by inappropriate activation of satellite cells in dystrophic muscle. (**A**) Sequence of events in nondystrophic muscle in which dystrophin negatively modulates satellite cell activation during muscle activity (yellow blocks outlined in black). (**B**) Pathogenic sequence of events in dystrophic muscle initiated by inappropriate activation of satellite cells during muscle activity (yellow blocks outlined in black) secondary to the absence of dystrophin-dependent modulation. Deleterious consequences are shown in light red blocks outlined in red.

## 2. Background

The hypothesis states that the absence of dystrophin leads to fiber death through unrestrained satellite cell activation that results in sustained myoblast proliferation and fusion of myoblast nuclei with the muscle membrane [16]. According to this proposal, the

direct effect of the absence of dystrophin is not muscle membrane structural breakdown, but instead a lack of control over the mechanism(s) by which muscle activity regulates satellite cell activation and fiber repair during passive stretch and concentric and eccentric exercise. Taken together with several studies showing that branched fibers, accumulated over time by chronically elevated levels of myoblast proliferation and fusion, are preferentially ruptured during muscle activity [17–21], this suggestion provides a new theory of dystrophic pathogenesis that merits serious consideration.

The idea that the lack of dystrophin directly results in unrestrained satellite cell activation, myoblast proliferation, and muscle membrane fusion is based on several observations showing that in vivo treatment with ActRIIB ligand trapping agents had quite unusual effects in an mdx muscle that exhibits a severely dystrophic phenotype, the mdx triangularis sterni (TS) muscle [15,16,22,23]. In nondystrophic muscle, inhibition, or blockade of myostatin signaling leads to muscle growth through two distinct mechanisms. Based upon the results of myostatin knockout studies, myostatin blockade in embryonic nondystrophic muscle increases myoblast proliferation and produces an increase in the number of fibers within the developing muscle (fiber hyperplasia), with variable effects on individual fiber size [24–27]. Sometime during embryonic development, this hyperplasic response to myostatin inhibition changes to one characterized by individual fiber growth (fiber hypertrophy), with no effects on the number of developing muscle fibers. In nondystrophic postnatal muscle, myostatin inhibition invariably produces only fiber hypertrophy with no effects on the number of muscle fibers [28–31]. In other words, the response of intramuscular nuclei (myonuclei and satellite cell nuclei) to myostatin inhibition changes during embryonic development from one of satellite cell activation and myoblast proliferation to one characterized solely by the activation of those signaling pathways that promote individual fiber growth by existing myonuclei.

In distinct contrast to results from postnatal nondystrophic muscle [28–31], selective inhibition of myostatin signaling with an ActRIIB ligand trapping agent (RAP-031, RAP-435) produced profound hyperplasia of both myonuclei and fiber cross sections (thus termed because of the presence of split fibers in dystrophic muscle), but had no effect on fiber size in the severely dystrophic [15] adult mdx TS muscle [16]. As expected, however, the treatment produced the opposite—significant fiber hypertrophy with no hyperplasia in age-matched nondystrophic mice [16]. These results directly indicate that dystrophin promotes satellite cell quiescence, and that the absence of the protein thereby leads to some degree of unrestrained postnatal satellite cell activation and myoblast proliferation and fusion that likely depends on muscle activation history. Consistent with this idea, examination of aging mdx TS muscle showed an uninterrupted and progressive hyperplasia of both myonuclei and fiber cross sections between 2 and 10 months that was not seen in nondystrophic preparations [15,16].

To assess whether the progressive hyperplasia observed in the mdx TS was secondary to prior fiber injury, Nielsen et al. [16] exhaustively compared distributions of fiber cross sectional area between untreated nondystrophic TS muscles at 2 and 4 months to age-matched vehicle and RAP-435 treated mdx TS muscles. Since injury of nondystrophic fibers induces the formation of a distinct population of smaller regenerating fibers [4], the hypothesis that fiber injury precedes regeneration in dystrophic muscle predicts that distributions of fiber cross sectional area in dystrophic muscle should exhibit a distinct population of smaller diameter fibers that are not seen in uninjured nondystrophic muscle. The results, however, showed that both nondystrophic and vehicle-treated mdx TS muscles exhibited rightward skewed distributions of fiber cross sectional area that were 90% identical, with no distinct subpopulations of fibers in either preparation. The mdx TS muscles did exhibit a reduction in the modal cross-sectional area, an effect which would be consistent with an enhanced rate of fiber generation associated with fiber splitting. Further stimulation of fiber generation by 3 months of myostatin inhibition (RAP-435) produced distributions of fiber cross sectional area that were 93% identical to the distributions from age-matched vehicle treated mdx mice, thus indicating that even a RAP-435-induced doubling of fiber

cross sections occurred without any evidence of fiber injury. These results are clearly inconsistent with the requirement that fiber injury precede satellite cell regeneration in dystrophic muscle (Figure 1), and indicate that the primary defect induced by the absence of dystrophin in the mdx TS is an in situ failure to properly regulate satellite cell activation and subsequent myoblast proliferation and fusion (Figure 2) [16].

A valid criticism to hypotheses challenging membrane structural mechanisms of dystrophic pathogenesis [12] has been posed by the question: "If there are no structural deficits in the membranes of dystrophic muscle fibers, why are dystrophic muscles more susceptible to the damaging effects of eccentric muscle contractions [32,33]?". The answer to this question has been obtained by a series of excellent studies determining the accumulation of split fibers and the susceptibility of split fibers to damage during disease progression [17–21]. First, these studies showed that younger mdx muscle does not show hypersensitivity to eccentric contraction damage, thus indicating that loss of dystrophin is not sufficient for contraction-induced membrane damage, which, of course, would be predicted by the membrane structural hypothesis [17]. Second, the degree of contraction-induced damage and tension loss during a series of eccentric contractions was positively correlated with age and the proportion of branched fibers in mdx muscles [17,21]. Third, tension recordings from isolated mdx fibers indicated that breakage selectively occurs at branch points that are structurally weaker than unbranched regions [18,20]. These results have been explained by a two-stage model in which abnormal ion channel function [12] and elevated ROS produce increases in intracellular $Ca^{2+}$ that induce muscle necrosis and secondarily induce muscle regeneration that ultimately leads to enhanced fiber branching and widespread fiber death [20,21].

However, the initial stage in this pathogenic sequence [20,21] is unnecessary if indeed fiber generation is directly accelerated by the absence of dystrophin (Figure 2) [16]. Furthermore, direct evidence for global $Ca^+$ accumulation in dystrophic muscle has been rather weak and quite controversial [12,34], while direct fluorimetric measurements indicate identical rates of resting $Ca^{2+}$ influx between intact mdx and nondystrophic adult muscle fibers [22,35]. One can conclude, therefore, that widespread increases in resting $Ca^{2+}$ influx are not pathogenic since they do not exist in severely dystrophic mdx muscle [22]. However, the presence of split fibers is directly pathogenic [17–21], and results showing that dystrophin promotes satellite cell quiescence [16] suggests a simple pathogenic model whereby some degree of unrestrained satellite cell activation, modulated by muscle activity, ultimately leads to the accumulation of structurally weak branched fibers (Figure 2) [21]. These branched fibers, initially observed in biopsies from patients with Duchenne muscular dystrophy [36], eventually rupture during normal muscle activity [17–21], and produce hypercontracted sarcoplasm, adjacent areas of empty sarcoplasm, and plasma membrane vacuolization that are all characteristic of the more terminal stages of fiber death in the severely dystrophic mdx TS muscle [15]. The infiltration of extracellular dyes, such as Evans Blue, is thus explained by selective entry at ruptured membrane branch points exhibiting hypercontraction [37], and not by more generalized entry through leaky membranes or through focal "delta lesions" [10] which have not been reproducibly observed in either human [13] or murine [14,15] dystrophic muscle. The presence of membrane vacuolization in regions adjacent to hypercontraction plugs [15] is entirely consistent with the selective entry of Evans Blue in hypercontracted regions [37] and provides further evidence that membrane disruption is solely a late-stage event in dystrophic pathogenesis [15].

## 3. Conclusions

Results demonstrating that exercise and training activate and induce the proliferation of satellite cells in nondystrophic muscle [2,3] imply that muscle activity regulates satellite cells nestled within the envelope between the basal lamina and the muscle membrane. The current proposal suggests that this regulation is partly mediated by dystrophin and the internal muscle membrane cytoskeleton (Figure 2). According to this proposal, the absence of dystrophin would result in some degree of unrestrained satellite cell activation

and subsequent myoblast proliferation and fusion that would depend upon muscle activity and patterns of muscle use. In the mdx mouse, progression of the disease appears to be more rapid and complete in the mdx TS muscle [15,22] than in the limb musculature, which is relatively spared and responds to myostatin inhibition with some degree of fiber hypertrophy [38–41]. The clear absence of fiber hypertrophy with myostatin inhibition in the severely dystrophic mdx TS [16] suggests that the pattern of use in this muscle promotes satellite cell activation and subsequent myoblast proliferation and fusion that is highly dysregulated. This dysregulation may be partly responsible for the clinical failure of myostatin inhibitors in trials with Duchenne muscular dystrophy patients [42], since the more severely dystrophic muscles in such patients would be expected to be morphologically and functionally worsened by a treatment which promotes unrestrained satellite cell activation, myonuclei and fiber hyperplasia, fiber splitting, and eventual fiber rupture (Figure 2) [16,21]. In my opinion, it is time to suspend efforts to explain a putative membrane structural role for dystrophin and redirect those efforts towards improving our understanding of satellite cell activation during muscle activity and the potential role of dystrophin in mediating or opposing that activation (Figure 2).

**Funding:** The authors cited research was supported by NIH grant R15AR055360, the Association Francaise contre les Myopathies (11832, 13980), Charleys Fund, the Warner Fermaturo Fund (ATSU), Missouri Department of Health Contract (ERS 186 010-01) and a Strategic Research Grant from AT Still University.

**Institutional Review Board Statement:** Not applicable.

**Informed Consent Statement:** Not applicable.

**Conflicts of Interest:** The authors declare no conflict of interest.

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
