# Peer review of "Does the Pathogenic Sequence of Skeletal Muscle Degeneration in Duchenne Muscular Dystrophy Begin and End with Unrestrained Satellite Cell Activation?"

_muscles, doi:10.3390/muscles1010008_

Round 1

Reviewer 1 Report

This is an interesting opinion paper from George Carlson who has presented results many from his own team which are in support of a new hypothesis for the pathophysiology of duchenne muscular dystrophy and the role of the protein mutated in this disease - dystrophin. It is refreshing to have a paper which will make people leave the general dogma to look at the physiopathologie of the mdx mouse in a different light it may also stimulate new areas of research on the role of satellite cell biology in muscle disease.

I have only a few very small comments.

Firstly, I find the term satellite cell regeneration a little strange and unconventional. The literature generally talks about satellite activation/proliferation and muscle regeneration. Can you really talk about satellite cells regenerating?

Secondly, there is a vast literature on myostatin in the mdx mouse as well as the LGMD dog and all of the studies show hypertrophy of the muscle fibres. The only time hyperplasia is observed is when myostatin is inhibited in the developing muscle. Therefore are the observations reported here really specific to the triangularis sterni (TS) muscle in the mouse or do they observe this in other muscles in their study?

Leaky membranes are reported throughout the literature using evans blue dye or calcium markers. Therefore it seems to difficult to exclude the fragile membrane theory.

Branched fibres are fascinating and not frequently discussed can the authors cite reports of this in DMD? 

Author Response

  1.  I have changed the wording throughout the text to conform to the appropriate definitions regarding satellite cell activation. Thank you very much for noticing this error.
  2. It is true that other studies have indicated some hypertrophy with myostatin blockade in dystrophic muscle. Although I did not cite the relevant work in the dog, I do cite the relevant work on limb muscles in the mdx mouse and indicate that the difference with the TS muscle may be the history of activation of this muscle which is much more severely affected than other mdx muscles. From the last paragraph: "According to this proposal, the absence of dystrophin would result in some degree of unrestrained satellite cell activation and subsequent myoblast proliferation and fusion that would depend upon muscle activity and patterns of muscle use. In the mdx mouse, progression of the disease appears to be more rapid and complete in the mdx TS muscle [15, 22] than in the limb musculature, which is relatively spared and responds to myostatin inhibition with some degree of fiber hypertrophy [38 - 41]. The clear absence of fiber hypertrophy with myostatin inhibition in the severely dystrophic mdx TS [16] suggests that the pattern of use in this muscle promotes satellite cell activation and subsequent myoblast proliferation and fusion that is highly dysregulated. "
  3. In their original paper describing Evans Blue infiltration, the authors explicitly indicate that "Evans Blue - stained muscle fibers were either hypercontracted or degenerating" (from abstract of Matsuda et al (1995)). I have therefore added the following: "The infiltration of extracellular dyes such as Evans Blue is thus explained by selective entry at ruptured membrane branch points exhibiting hypercontraction [37], and not by more generalized entry through leaky membranes or through focal "delta lesions" [10], which have not been reproducibly observed in either human [13] or murine [14, 15] dystrophic muscle. The presence of membrane vacuolization in regions adjacent to hypercontraction plugs [15] is entirely consistent with the selective entry of Evans Blue in hypercontracted regions [37] and provides further evidence that membrane disruption is solely a late stage event in dystrophic pathogenesis [15]. "
  4.  The initial reference demonstrating branched fibers in Duchenne muscular dystrophy has been added. "These branched fibers, initially observed in biopsies from patients with Duchenne muscular dystrophy [36], ......... 
  5. Thank you for your suggestions.  

Reviewer 2 Report

This is a well written and thought provoking manuscript. I applaud the author for the call to arms to rethink how we phrase questions about muscular dystrophy.

The main problem with the overall hypothesis that unprovoked stem cell proliferation explains DMD is that C. elegans lack satellite cells (and an inflammatory system) and yet C. elegans display DMD that can be treated with prednisone (e.g. they are a valid clinical model without satellite cells).

This observation of lack of satellite cells in a valid clinical model must be incorporated into the article in order for it to be scientifically sound. Otherwise we risk more research based on an incompletely considered hypothesis (which is what the author is arguing against in the first place).

Author Response

Thank you very much for your encouraging and generous comments. However, I must respectfully disagree that c. elegans  is a valid clinical model for Duchenne muscular dystrophy precisely because it lacks satellite cells. I also respectfully disagree with the comment that since prednisone can be used to treat mutant c elegans, c elegans therefore displays DMD that can be treated with prednisone and is therefore a "valid clinical model" for DMD that lacks satellite cells. DMD is not defined as a disease that responds favorably to prednisone, The phenotype of dystrophy in c elegans is distinct from the phenotype in Duchenne muscular dystrophy (and the mdx mouse) which is primarily characterized by chronic degeneration and regeneration (requiring satellite cells).  Results from c elegans could indeed be useful for understanding some potential functions of dystrophin in human muscle. However, since the mechanisms of satellite cell activation are central to this opinion paper (and central to pathogenesis in DMD), I do not see the relevance of including discussion from this particular model that lacks satellite cells and therefore cannot yield information regarding the behavior of satellite cells in mammalian muscle.  

Round 2

Reviewer 2 Report

While I am disappointed that the author chose to suggest that because C. elegans lack satellite cells that they are therefore not a relevant clinical model I accept that many people do not view the worm as a valid clinical model. I remain concerned that raising a hypothesis that clearly isn't supported by various lines of evidence can be dangerous. However, the addition of Figure 1 is sufficient to address my concern that it is clear that everything does not begin and end with satellite cells. Additionally, I continue to agree that new lines of research are needed for DMD and that this article is a positive step in that direction.